# NUMERIC ENCODING OPTIONS WITH AUTOMUNGE

## ABSTRACT

Mainstream practice in machine learning with tabular data may take for granted that any feature engineering beyond scaling for numeric sets is superfluous in context of deep neural networks. This paper will offer arguments for potential benefits of extended encodings of numeric streams in deep learning by way of a survey of options for numeric transformations as available in the Automunge open source python library platform for tabular data pipelines, where transformations may be applied to distinct columns in "family tree" sets with generations and branches of derivations. Automunge transformation options include normalization, binning, noise injection, derivatives, and more. The aggregation of these methods into family tree sets of transformations are demonstrated for use to present numeric features to machine learning in multiple configurations of varying information content, as may be applied to encode numeric sets of unknown interpretation. Experiments demonstrate the realization of a novel generalized solution to data augmentation by noise injection for tabular learning, as may materially benefit model performance in applications with underserved training data.

## 1 INTRODUCTION

Of the various modalities of machine learning application (e.g. images, language, audio, etc.) tabular data, aka structured data, as may comprise tables of feature set columns and collected sample rows, in my experience does not command as much attention from the research community, for which I speculate may be partly attributed to the general non-uniformity across manifestations precluding the conventions of most other modalities for representative benchmarks and availability of pre-trained architectures as could be adapted with fine-tuning to practical applications. That is not to say that tabular data lacks points of uniformity across data sets, for at its core the various feature sets can at a high level be grouped into just two primary types: numeric and categoric. It was the focus of a recent paper by this author (Author, 2020) to explore methods of preparing categoric sets for machine learning as are available in the Automunge open source python library platform for tabular data pipelines. This paper will give similar treatment for methods to prepare numeric feature sets for machine learning.

Of course it would be an oversimplification to characterize "numeric feature sets" as a sufficient descriptor alone to represent the wide amount of diversity as may be found between different such instances. Numeric could be referring to integers, floats, or combinations thereof. The set of entries could be bounded, the potential range of entries could be bounded on the left, right, or both sides, the distribution of values could be thin or fat tailed, single or multi-modal. The order of samples could be independent or sequential. In some cases the values could themselves be an encoded representation of a categoric feature.

Beyond the potential diversity found within our numeric features, another source of diversity could be considered based on relationships between multiple feature sets. For example one feature could be independent of the others, could contain full or partial redundancy with one or more other variables by correlation, or in the case of sequential data there could even be causal relationships between variables across time steps.

The primary focus of transformations to be discussed in this paper will not take into account variable interdependencies, and will instead operate under the assumption that the training operation of a downstream learning algorithm may be more suitable for the efficient interpretation of such interdependencies, as the convention for Automunge is that data transformations (and in some cases

sets of transformations) are to be directed for application to a distinct feature set as input. In many cases the basis for these transformations will be properties derived from the received feature in a designated "train" set (as may be passed to the automunge(.) function) for subsequent application on a consistent basis to a designated "test" set (as may be passed to the postmunge(.) function).

## 2 NORMALIZATIONS

A common practice for preprocessing numeric feature sets for the application of neural networks is to apply a normalization operation in which received values are centered and scaled based on properties of the data. By conversion to comparable scale between features, backpropagation may have an easier time navigating the fitness landscape rather than overweighting to higher magnitude inputs (Ng, 2011). Table 1 surveys a few normalization operations as available in Automunge.

Table 1: Normalizations

| Type | ID | Formula | Scaling |
|------|------|---------|---------|
| z-score | 'nmbr' | $(x_i - \mu)/\sigma$ | scaled to sigma 1 and mu 0 |
| min-max | 'mnmx' | $(x_i - min)/(max - min)$ | scaled to unit interval |
| mean | 'mean' | $(x_i - mean)/(max - min)$ | scaled and centered to mean |
| MAD | 'MAD3' | $(x_i - max)/(MAD)$ | scaled by median absolute deviation |
| lognorm | 'lgnm' | $\ln(x_i) \rightarrow (x_i - \mu)/\sigma$ | log-normal scaled to Gaussian |

Upon inspection a few points of differentiation become evident. The choice of denominator can be material to the result, for while both (max - min) and standard deviation can have the result of shrinking or enlarging the values to fall within a more uniform range, the (max - min) variety has more of a known returned range for the output that is independent of the feature set distribution properties, thus allowing us to ensure all of the min-max returned values are non-negative for instance, as may be a pre-requisite for some kinds of algorithms. Of course this known range of output relies on the assumption that the range of values in subsequent test sets will correspond to the train set properties that serve as a basis - to allow a user to prevent these type of outliers from interfering with downstream applications, Automunge allows a user to pass parameters to the transformation functions, such as to activate floors or caps on the returned range.

An easy to overlook outcome of the shifting and/or centering of the returned range by way of the subtraction operation in the numerator is a loss of the original zero point, as for example with z-score normalization the returned zero point is shifted to coincide with the original mean. It is the opinion of this author that such re-centering of the data may not always be a trivial trade-off. Consider the special properties of the number 0 in mathematics, including multiplicative properties at/above/below. By shifting the original zero point we are presenting a (small) obstacle to the training operation in order to relearn this point. Perhaps more importantly, further trade-offs include the interpretability of the returned data.

Automunge thus offers a novel form of normalization, available in our library as 'retn' (standing for "retain"), that bases the formula applied to scale data on the range of values found within the train set, with the result of scaling the data within a similar range as some of those demonstrated above while also retaining the zero point and thus the +/- sign of all received data.

Table 2: Retain Normalization ('retn')

| Min | Max | Formula | Returned Min | Returned Max |
|------|------|---------|--------------|--------------|
| $\leq 0$ | $\geq 0$ | $x_i/(max - min)$ | $min/(max - min)$ | $max/(max - min)$ |
| $> 0$ | $> 0$ | $(x_i - min)/(max - min)$ | 0 | 1 |
| $< 0$ | $< 0$ | $(x_i - max)/(max - min)$ | -1 | 0 |

## 3 Transformations

In many cases the application of a normalization procedure may be preceded by one or more types of data transformations applied to the received numeric set. Examples of data transformations could include basic mathematic operators like + - * /, log transforms, raising to a power, absolute values, etc. In some cases the transformations may also be tailored to the properties of the train set, for example with a Box-Cox power law transformation (Box & Cox, 1964).

In the Automunge library, the order of such sets of transformations, as may be applied to a distinct source column, and in some cases which may include generations and branches of derivations, are specified by way of transformation category entries to a set of "family tree" primitives [Table 3] (Author, 2020) for a root transformation category, and where a transformation category entry may be associated with one or more transformation functions intended for application to corresponding train and/or test set feature columns, potentially including custom transformation functions which may be defined with minimal requirements of simple data structures. Such root categories may be pre-defined in the Automunge library of transformations or may be custom configured by a user in entries to a "transformdict" data structure.

Table 3: Family Tree Primitives

| Primitive | Upstream / Downstream | Applied to Generation | Column Action | Downstream Offspring |
|---|---|---|---|---|
| parents | upstream | first | replace | yes |
| siblings | upstream | first | supplement | yes |
| auntsuncles | upstream | first | replace | no |
| cousins | upstream | first | supplement | no |
| children | downstream parents | offspring | replace | yes |
| niecesnephews | downstream siblings | offspring | supplement | yes |
| coworkers | downstream auntsuncles | offspring | replace | no |
| friends | downstream cousins | offspring | supplement | no |

The convention for transformation functions in the Automunge library is that any kind of function accepts any kind of data, and in cases where an invalid entry is returned, for example when dividing by zero or taking a square root of a negative number, such entry may serve as a target for missing data infill, with infill methods that may be applied to a column from a library of infill options - including "ML infill" in which random forest models (Breiman, 2001) are used to predict infill based on properties of the train set. To facilitate the application of infill, transformation categories used as root categories are specified with a classification for the types of data that will be considered valid input, as for example may be non-negative numeric, non-zero numeric, integer numeric, etc. These "NArowtype" classifications are populated in the same "processdict" data structure used to assign transformation functions to a transformation category.

## 4 Bins and Grainings

In most cases the transformations considered in the preceding section maintained full information retention of the received data, such that with returned sets the form of the input data can be recovered with an inversion operation (as is available in the Automunge library). For binning transformations, there may instead be a type of coarse graining of the feature set to aggregate buckets of entries into a categoric representation. Automunge offers a wide range of options for numeric binning [Table 4]. Bins may be aggregated to either supplement or replace received numeric sets.

For each binning operation, transformation category options are available to return the categoric encoding as a one-hot encoding, ordinal integer encoding, or binary encoding in which distinct categories may be represented by multiple simultaneous activations. This is partly motivated by different conventions of various libraries for accepting input to an entity embedding layer (Guo & Berkhahn, 2016) as may be applied to the returned categoric encoding in a downstream training operation.

Table 4: Binning Transformation Category Options

| Transform | One-Hot | Ordinal | Binary | Parameters |
|---|---|---|---|---|
| Number of standard deviations from the mean | 'bins' | 'bsor' | 'bsbn' | 'bincount' |
| Powers of ten | 'pwrs' | 'pwor' | 'pwbn' | - |
| Powers of ten (with support for negative) | 'pwr2' | 'por2' | 'por3' | - |
| Fixed width bins | 'bnwd' | 'bnwo' | 'bnwb' | 'width' |
| Equal population bins | 'bnep' | 'bneo' | 'bneb' | 'bincount' |
| User specified bins (first/last unconstrained) | 'bkt1' | 'bkt3' | 'bkb3' | 'buckets' |
| User specified bins (first/last bounded) | 'bkt2' | 'bkt4' | 'bkb4' | 'buckets' |

## 5 NOISE INJECTION

For most cases in the Automunge library, transformations applied to a train set feature set are applied to the corresponding test set feature set using the same basis, such that if the same data is received for both train and test sets, the same form will be returned (a useful point for validations). The noise injection options are a little different in that such injections may be intended just for the train data but not the corresponding test data.

The rationale behind noise injections were first to support differential privacy considerations (Dwork et al., 2006). Other potential uses of noise injections could be to perturb the model training, facilitating diversity between models as may be beneficial in the aggregation of ensembles (Dietterich, 2000) or as a source of training data augmentation (Perez & Wang, 2017). The experiments detailed below will suggest a material model performance benefit from data augmentation by noise injection in cases of underserved training data, which we believe is a novel innovation for tabular learning.

The options available for noise injection are generally derived by way of aggregations of transformation category entries to family tree primitives, as noise injection may be applied downstream to a normalization or categoric encoding. (The convention for transformation functions is that they receive input of a single target column, so transformations performed downstream of a categoric encoding should be fed an ordinal input.) The library includes distinct noise injection family tree aggregations tailored to operation of several different types of received normalizations, as may rely on a known range or scale of input, or applied preceding different types of categoric encodings.

Table 5: Numeric Noise Injections

| Root Category | Normalization | Noise Type | Parameters |
|---|---|---|---|
| 'DPnb' | 'nmbr' | Gaussian w/ Bernoulli ratio | 'mu' / 'sigma' / 'flip_prob' |
| 'DPmm' | 'mnmx' | scaled Gaussian w/ Bernoulli ratio | 'mu' / 'sigma' / 'flip_prob' |
| 'DPrt' | 'retn' | scaled Gaussian w/ Bernoulli ratio | 'mu' / 'sigma' / 'flip_prob' |

Table 6: Categoric Noise Injections

| Root Category | Encoding | Noise Type | Parameters |
|---|---|---|---|
| 'DPbn' | 'bnry' (boolean) | Bernoulli flip | 'flip_prob' |
| 'DPod' | 'ord3' (ordinal) | Bernoulli flip to random activation | 'flip_prob' |
| 'DPoh' | 'onht' (one-hot) | Bernoulli flip to random activation | 'flip_prob' |
| 'DP10' | '1010' (binary) | Bernoulli flip to random activation set | 'flip_prob' |

Numeric noise injections [Table 5, Fig 1] are derived from a Gaussian source with configurable parameters. For noise intended to sets with a fixed range of values such as DPmm, although the

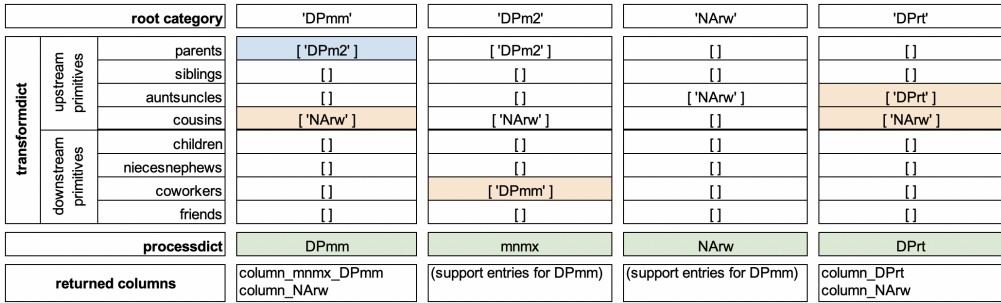

| | | root category | 'DPmm' | 'DPm2' | 'NArw' | 'DPrt' |
|---|---|---|---|---|---|---|
| **transformdict** | upstream primitives | parents | [ 'DPm2' ] | [ 'DPm2' ] | [] | [] |
| | | siblings | [] | [] | [] | [] |
| | | auntsuncles | [] | [] | [ 'NArw' ] | [ 'DPrt' ] |
| | | cousins | [ 'NArw' ] | [ 'NArw' ] | [] | [ 'NArw' ] |
| | downstream primitives | children | [] | [] | [] | [] |
| | | niecesnephews | [] | [] | [] | [] |
| | | coworkers | [] | [ 'DPmm' ] | [] | [] |
| | | friends | [] | [] | [] | [] |
| | | **processdict** | DPmm | mnmx | NArw | DPrt |
| | | **returned columns** | column_mnmx_DPmm column_NArw | (support entries for DPmm) | (support entries for DPmm) | column_DPrt column_NArw |

Figure 1: 'DPmm' and 'DPrt' family trees

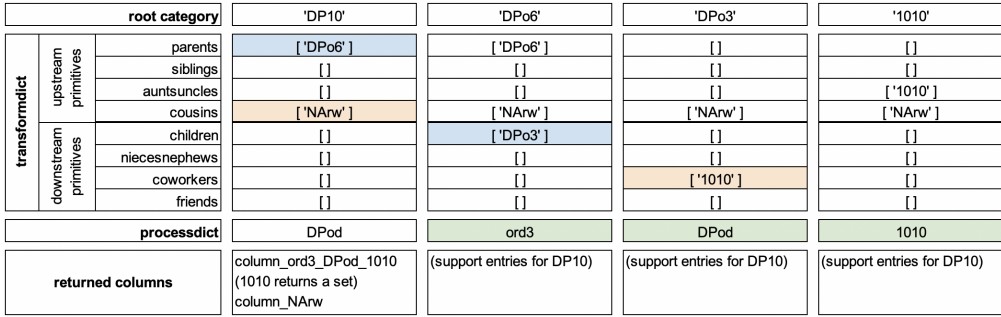

| | | root category | 'DP10' | 'DPo6' | 'DPo3' | '1010' |
|---|---|---|---|---|---|---|
| **transformdict** | upstream primitives | parents | [ 'DPo6' ] | [ 'DPo6' ] | [] | [] |
| | | siblings | [] | [] | [] | [] |
| | | auntsuncles | [] | [] | [] | [ '1010' ] |
| | | cousins | [ 'NArw' ] | [ 'NArw' ] | [ 'NArw' ] | [ 'NArw' ] |
| | downstream primitives | children | [] | [ 'DPo3' ] | [] | [] |
| | | niecesnephews | [] | [] | [] | [] |
| | | coworkers | [] | [] | [ '1010' ] | [] |
| | | friends | [] | [] | [] | [] |
| | | **processdict** | DPod | ord3 | DPod | 1010 |
| | | **returned columns** | column_ord3_DPod_1010 (1010 returns a set) column_NArw | (support entries for DP10) | (support entries for DP10) | (support entries for DP10) |

Figure 2: 'DP10' family trees

noise source as implemented is Gaussian, the application is capped from extreme outliers at half of range (e.g. +/- 0.5) and based on whether an input entry is above or below the midpoint, positive or negative noise respectively is scaled to ensure maintained original range in returned data based on values of input entry. Parameters are also accepted to indicate what ratio of input will receive injection. Similar options are available for Laplace distribution noise profiles.

To clarify how family tree primitives come into play, for the 'DPmm' root category in Fig 1, when 'DPmm' is applied as a root category to a source column with header 'column', the upstream primitive entries are inspected, which include a parents entry of the 'DPm2' transformation category and a cousins entry of the 'NArw' transformation category. For the 'NArw' upstream entry to the cousins primitive, the 'NArw' transformation category is associated with a NArw transformation function (a function aggregating boolean activations indicating presence of infill) per the processdict entry associated with the 'NArw' root category, which transformation function returns the column 'column_NArw' with the suffix appender '_NArw' logging the transformation function applied. Because cousins is a primitive without offspring no further generations are inspected in the 'NArw' family tree downstream primitives. For the upstream parents primitive entry 'DPm2', the 'DPm2' processdict entry has a transformation function of mnmx (min-max scaling), which is applied and logged by the '_mnmx' suffix appender, and since parents is a primitive with offspring the downstream primitives in the 'DPm2' family tree are inspected where a coworkers entry of 'DPmm' transformation category is found which is associated with a DPmm transformation function (noise injection corresponding to range 0-1) based on the 'DPmm' root category processdict entry. Because coworkers is a replacement primitive the column configuration with header 'column_mnmx' is not retained for the returned set. The returned column originating from the DPmm transformation function has header logging the two applied transformation functions as 'column_mnmx_DPmm'. Please note that column retention, as may be impacted by the application of downstream replacement primitive entries, is signaled in these diagrams by the color coding between orange and blue, and applied transformation functions are shaded as green. Transformation category entries of primitives that are not inspected are left without shading. Also please note that the processdict entries here are an abstraction for the set of corresponding transformation functions that may be directed to train and/or test set feature sets.

The application of noise to categoric encodings [Table 6, Fig 2] is a little simpler, where a given ratio of entries in a categoric feature set are flipped to one of the other activations, between which have a uniform probability of replacement (including possibility of original entry retention). For the 'DP10' root category example shown in Fig 2, this is achieved by first applying an ordinal encoding by ord3 transformation function associated with a 'DPo6' transformation category entry to the parents upstream primitive of the 'DP10' root category, then a noise injection by the DPod transformation function associated with a 'DPo3' transformation category to the downstream children primitive associated with the 'DPo6' root category (applied since parents is a primitive with offspring), followed by a binary encoding by the 1010 transformation function associated with the '1010' transformation category entry to the downstream coworkers primitive associated with the 'DPo3' root category (applied since children is a primitive with offspring). Here the intermediate stages of derivations associated with columns 'column_ord3' and 'column_ord3_DPod' are not retained in the returned set since children and coworkers are replacement primitives.

## 6 SEQUENTIAL DATA

For numeric features in which the order of samples carry some significance, e.g. for time series data, some additional options are available to extract structure from relationships between time steps (these methods may benefit from a convention that time deltas between measurements are at or nearly constant). The theory is that for sequential machine learning applications, as may make use of recurrence, convolution, or attention mechanisms, there may be benefit to supplementing feature streams with properties carried forward from prior time steps, where this type of operation may be particularly beneficial when there is some known cyclic property inherent in the application, e.g. for scheduled trading hours or quarterly reports.

Included in the Automunge library are sequential transforms to supplement sequential streams with proxies for derivatives by returning deltas between an entry and some desired time step prior by way of the 'dxdt' family of transforms [Fig 3]. Such application may be applied once as a proxy for velocity, and may also be run multiple times upon that output as proxies for higher order derivatives. A variant on this operation may, instead of taking point-wise deltas, return deltas between averages of sets of points, which has the effect of smoothing or de-noising the data. The outputs of these operations may each be normalized such as with the 'retn' normalization for sign retention.

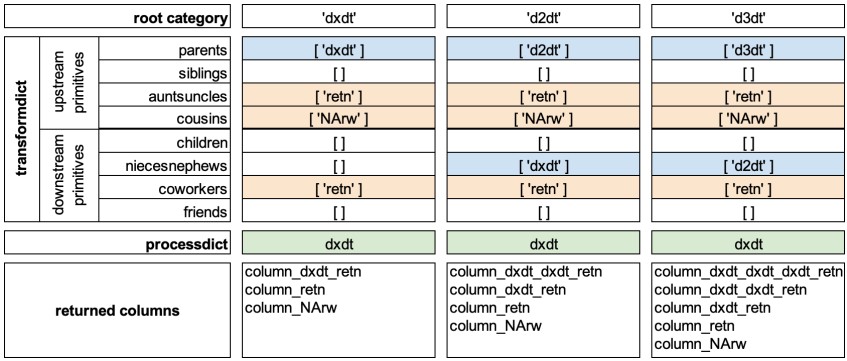

Figure 3: 'dxdt' family trees

## 7 INTEGER SETS

Integer feature sets of unknown origin may present a particular challenge for automated encodings, as these may be associated with a diverse set of interpretations, and may originate from continuous variables, counters, discrete relational variables (e.g. small/medium/large), or possibly even an ordinal categoric encoding (Stevens et al., 2020). The Automunge philosophy for these kind of ambiguities is to simply redundantly encode in a manner suitable for each [Table 7, Fig 4], defering to a training operation for determining relevancy.

Table 7: Integer Encoding Options

| Transform | Type | Useful For |
|---|---|---|
| 'retn' | Retain normalization | Continuous variables |
| 'pwr2' | Order of magnitude bins | Continuous variables |
| 'ordl' | Ordinal | Ordered categoric |
| '1010' | Binary | Discrete categoric |
| 'dxdt' | Sequential | Cumulative sequential |
| 'ord3_mnmx' | Frequency sorted ordinal followed by min-max | Scaled metric for ranking entry frequency |

| | | root category | 'ntgr' | 'ntg2' | 'ntg3' | 'mnmx' |
|---|---|---|---|---|---|---|
| transformdict | upstream primitives | parents | [ 'ntgr' ] | [ 'ntg2' ] | [ 'ntg2' ] | [ ] |
| | | siblings | [ ] | [ ] | [ ] | [ ] |
| | | auntsuncles | [ 'retn', '1010', 'ordl' ] | [ 'retn', '1010', 'ordl', 'pwr2' ] | [ 'retn', 'ordl', 'por2' ] | [ 'mnmx' ] |
| | | cousins | [ 'NArw' ] | [ 'NArw' ] | [ 'NArw' ] | [ 'NArw' ] |
| | downstream primitives | children | [ ] | [ ] | [ ] | [ ] |
| | | niecesnephews | [ ] | [ ] | [ ] | [ ] |
| | | coworkers | [ 'mnmx' ] | [ 'mnmx' ] | [ 'mnmx' ] | [ ] |
| | | friends | [ ] | [ ] | [ ] | [ ] |
| | | processdict | ord3 | ord3 | ord3 | mnmx |
| | | returned columns | column_ord3_mnmx column_retn column_1010 (set) column_ordl column_NArw | column_ord3_mnmx column_retn column_1010 (set) column_ordl column_pwr2 (set) column_NArw | column_ord3_mnmx column_retn column_ordl column_por2 column_NArw | (support entries for ntgr) |

Figure 4: 'ntgr' family trees

## 8 EXPERIMENTS

Some experiments were run to evaluate impact of various normalizations and transformations. The Higgs data set (Baldi et al., 2014) was selected based on scale and predominantly numeric feature types. The Higgs application is tabular data for binary classification sourced from high energy physics domain to evaluate subatomic particle interactions to distinguish between background noise and traces of Higgs boson interactions. The origin paper noted some details of architectures (5 layers with 300 nodes in each) that served as basis for the experiments, and we used a 0.6M size validation set out of the 8.8M samples. A departure was made from the origin paper in use of a phased learning rate with the fastai (Howard & Gugger, 2020) "fit_one_cycle" learner for tabular data, partly for convenience along with time and resource constraints. Since the interest was primarily to evaluate relative performance between different preprocessing methods, no significant attempt was made to train models to maximum performance or to venture beyond double descent, instead the primary tuning aspect was selecting an epoch depth beyond which additional did not have improved validation metrics. The experiments were repeated with different size subsets of the training data, including full data set, 5% samples, and 0.25% samples to represent scenarios with underserved training data. Learning rates deferred to the fastai default finder, and the evaluation metric was selected as area under the receiver operating characteristic curve (ROC AUC) to be consistent with the origin demonstration.

The experiments applied Automunge to preprocess the feature sets, in each case applying uniform transformation types between features although with basis fit to properties of each respective column, and with missing data infill by way of the Automunge adjacent cell infill option. The types of transformations were selected to demonstrate impact of a few representative variants noted in this writeup, and included a base scenario with no feature scaling applied (just raw numeric data), along with a scenario for z-score normalization, retain normalization, retain normalization supplemented by standard deviation bins, two scenarios of retain normalization with noise injection (with noise injected to all entries or to 3% of entries in each feature), and finally a data augmentation with retain normalized data supplemented by doubling the number of samples to include retain with partial

Table 8: Higgs Data Normalization Scenarios (AUC metric // compared to raw data)

| | Raw Data | Z-Score | Retain | Retain with Bins | Retain w/ Noise full | Retain w/ Noise partial | Retain w/ Augment |
|---|---|---|---|---|---|---|---|
| **full data** 42 epochs | 0.8670 - | 0.8668 (0.0002) | 0.8667 (0.0003) | 0.8660 (0.0010) | 0.8268 (0.0402) | 0.8624 (0.0046) | **0.8667** **(0.0003)** |
| **5% data** 14 epochs | 0.8436 - | 0.8432 (0.0004) | 0.8431 (0.0005) | 0.8433 (0.0003) | 0.7679 (0.0757) | 0.8388 (0.0048) | **0.8453** **0.0017** |
| **0.25% data** 3 epochs | 0.7715 - | 0.7716 0.0001 | 0.7716 0.0001 | 0.7718 0.0003 | 0.7052 (0.0663) | 0.7668 (0.0047) | **0.7821** **0.0106** |

noise injection. Both of the noise injections applied Gaussian noise with standard deviation of 0.03 and with scaling to maintain the fixed range of the received data.

The findings of the experiments are summarized in Table 8 by way of reporting the final validation set metrics and performance delta from the raw data scenario. The AUC metric shown compares to the origin paper's reported 88%, which was based on a reported 200-1,000 epochs compared to this experiment's 3-42 so there is some dissimilarity in results, which can also partly be attributed to the phased learning rate schedule applied by fastai. The AUC metrics shown are averages of repetition counts detailed in Appendix G.

The scenarios for raw data, z-score, and retain normalization achieved similar results. Part of the impact of supplementing the normalized data with standard deviation bins was a small increase of training time. As expected the noise injection had a dampening effect on the model accuracy, although the partial injection had only a small penalty in comparison to the retain normalization without injection. Data augmentation improved the results, sufficiently to draw the conclusion of material benefit for scenarios with underserved training data. We believe the realization of a novel generalized solution to data augmentation by noise injection for tabular learning is a significant finding.

The experiment with the full data set was repeated to demonstrate impact to a linear model:

Table 9: Higgs Data Normalization Scenarios (Support Vector Classifier with Linear kernel)

| | Raw Data | Z-Score | Retain | Retain with Bins | Retain w/ Noise full | Retain w/ Noise partial | Retain w/ Augment partial |
|---|---|---|---|---|---|---|---|
| **Accuracy** | 0.6410 - | 0.6410 - | 0.6410 - | **0.6821** **0.0411** | 0.6201 (0.0209) | 0.6393 (0.0018) | 0.6402 (0.0008) |

## 9 DISCUSSION

I would offer first that any consideration around benefits of feature engineering should distinguish first by scale of data available for training. When approaching big data scale input with infinite computational resources there may be less of a case to be made for much beyond basic normalized input. The nuance comes into play when we are targeting applications with real world constraints. Although deep over-parameterized models may still be applied to target applications with under-represented data (Olson et al., 2018), such methods carry overheads. It is one of the premises of

the Automunge library that supplementing our data streams with redundant features in multiple configurations may lower the bar to efficient extraction of inter-variable relationships.

My experience is that the machine learning community has by large become somewhat dismissive of any kind of feature engineering, I expect partly owed to such works as Deep Learning (Goodfellow et al., 2016) which offers that deep learning has supplanted the need for such effort. I would offer in retort that there are different kinds of feature engineering to consider. Those methods as may apply explorations and optimizations between inter-variable relationships I agree are largely redundant of what may be efficiently derived through backpropagation and this is part of the reason why Automunge hasn't ventured into this territory. But the types of data manipulations as may be used to supplement numeric features with multiple configurations are computationally efficient, and need not require sophisticated optimizations to apply. The primary cost of such supplements are the memory overheads as the data set size is expanded.

Further, variations on feature composition, such as by variations in information content and variations on injected noise, may serve as a useful source of model perturbations in ensemble assemblies. Noise injection may be of benefit for anonymizing sensitive data for purposes of differential privacy, and may also serve as a source of training data augmentation, similar to how for convolutional networks training data images may be duplicated with artificial variations (Perez & Wang, 2017).

I'll close by noting an interesting paper I saw at NeurIPS last year, "SGD on Neural Networks Learn Functions of Increasing Complexity" by Nakkiran et al. (2019), in which the authors found that SGD behaves as a linear model in early epochs, and importantly that characteristics of such linear models are retained even in the later stages of training. Put differently, any steps that may be made to ensure that our models may efficiently extract properties even in early stages, before deep learning can do its magic, are not immaterial to the final performance.

Table 10: Summary of Use Cases

| Desciption | Examples | Use Cases |
|---|---|---|
| Z-Score Normalization | nmbr | general use, when distribution is unknown |
| Min-Max Scaling | mnmx | when a fixed range is desired, e.g. nonnegative |
| Retain Normalization | retn | when interpretability is desired, sign retention |
| Bins and Grainings | bins, pwrs, bnep | to aggregate sets into buckets, such as to supplement normalized data, helpful with linear models |
| Noise Injection | DPnb, DPmm, DPrt, DPod, DP10 | useful for data set augmentation, especially with underserved training data, also for differential privacy |
| Sequential | dxdt, d2dt, d3dt | useful for time-series data, supplements normalized sets with proxies for derivatives |
| Integer Sets | ntgr, ntg2, ntg3 | to redundantly encode integer sets of unknown interpretation |

ACKNOWLEDGMENTS

A thank you is owed to Alice Zheng and Amanda Casari whose 2018 book "Feature Engineering for Machine Learning" served as a helpful reference as I began to explore the practice of feature engineering. Thanks to Stack Overflow, Python, PyPI, GitHub, Colaboratory, Anaconda, and Jupyter. Special thanks to Scikit-Learn, Numpy, Scipy Stats, and Pandas.

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

APPENDIX

## A FUNCTION CALL DEMONSTRATIONS

Automunge is available for pip install:

```
pip install Automunge
```

Or to upgrade (we currently roll out upgrades fairly frequently):

```
pip install Automunge --upgrade
```

Once installed, run this in local session to initialize:

```
from Automunge import Automunger
am = Automunger.AutoMunge()
```

Then, assuming we want to prepare a train set df_train for ML, can apply default parameters as:

```
train, trainID, labels, \
validation1, validationID1, validationlabels1, \
validation2, validationID2, validationlabels2, \
test, testID, testlabels, \
labelsencoding_dict, finalcolumns_train, finalcolumns_test, \
featureimportance, postprocess_dict = \
am.automunge(df_train)
```

Note that if our df_train set included a labels column, we should designate the column header with the labels_column parameter. Or likewise we can designate any ID columns with the trainID_column parameter.

The returned postprocess_dict should be saved such as with pickle.

We can then consistently prepare subsequent test data df_test in postmunge(.):

```
test, testID, testlabels, \
labelsencoding_dict, postreports_dict \
= am.postmunge(postprocess_dict, df_test)
```

I find it helps to just copy and paste the full range of parameters for reference:

```
train, trainID, labels, \
validation1, validationID1, validationlabels1, \
validation2, validationID2, validationlabels2, \
test, testID, testlabels, \
labelsencoding_dict, finalcolumns_train, finalcolumns_test, \
featureimportance, postprocess_dict = \
am.automunge(df_train, df_test=False, labels_column=False,
  trainID_column=False, testID_column=False, valpercent1=.0,
  valpercent2=.0, floatprecision=32, shuffletrain=True,
  TrainLabelFreqLevel=False, powertransform=False,
  binstransform=False, MLinfill=False, infilliterate=1,
  randomseed=42, eval_ratio=.5, LabelSmoothing_train=False,
  LabelSmoothing_test=False, LabelSmoothing_val=False, LSfit=False,
  numbercategoryheuristic=63, pandasoutput=False,
  NArw_marker=False, featureselection=False, featurepct=1.0,
  featuremetric=0.0, featuremethod='default', Binary=False,
  PCAn_components=False, PCAexcl=[], excl_suffix=False,
  ML_cmnd = {'MLinfill_type':'default',
          'MLinfill_cmnd':{'RandomForestClassifier':{},
```

```
                                'RandomForestRegressor':{}},
            'PCA_type':'default', 'PCA_cmnd':{}},
 assigncat = {
 'nmbr':[], 'retn':[], 'mnmx':[], 'mean':[], 'MAD3':[], 'lgnm':[],
 'bins':[], 'bsor':[], 'pwrs':[], 'pwr2':[], 'por2':[], 'bxcx':[],
 'addd':[], 'sbtr':[], 'mltp':[], 'divd':[],
 'log0':[], 'log1':[], 'logn':[], 'sqrt':[], 'rais':[], 'absl':[],
 'bnwd':[], 'bnwK':[], 'bnwM':[], 'bnwo':[], 'bnKo':[], 'bnMo':[],
 'bnep':[], 'bne7':[], 'bne9':[], 'bneo':[], 'bn7o':[], 'bn9o':[],
 'bkt1':[], 'bkt2':[], 'bkt3':[], 'bkt4':[],
 'nbr2':[], 'nbr3':[], 'MADn':[], 'MAD2':[], 'tlbn':[],
 'mnm2':[], 'mnm3':[], 'mnm4':[], 'mnm5':[], 'mnm6':[],
 'ntgr':[], 'ntg2':[], 'ntg3':[], 'mea2':[], 'mea3':[], 'bxc2':[],
 'dxdt':[], 'd2dt':[], 'd3dt':[], 'dxd2':[], 'd2d2':[], 'd3d2':[],
 'nmdx':[], 'nmd2':[], 'nmd3':[], 'mmdx':[], 'mmd2':[], 'mmd3':[],
 'shft':[], 'shf2':[], 'shf3':[], 'shf4':[], 'shf7':[], 'shf8':[],
 'bnry':[], 'onht':[], 'text':[], 'txt2':[], '1010':[], 'or10':[],
 'ordl':[], 'ord2':[], 'ord3':[], 'ord4':[], 'om10':[], 'mmor':[],
 'Unht':[], 'Utxt':[], 'Utx2':[], 'Uor3':[], 'Uor6':[], 'U101':[],
 'splt':[], 'spl2':[], 'spl5':[], 'sp15':[], 'sp19':[], 'sbst':[],
 'spl8':[], 'spl9':[], 'sp10':[], 'sp16':[], 'sp20':[], 'sbs2':[],
 'srch':[], 'src2':[], 'src4':[], 'strn':[], 'lngt':[], 'aggt':[],
 'nmrc':[], 'nmr2':[], 'nmcm':[], 'nmc2':[], 'nmEU':[], 'nmE2':[],
 'nmr7':[], 'nmr8':[], 'nmc7':[], 'nmc8':[], 'nmE7':[], 'nmE8':[],
 'ors2':[], 'ors5':[], 'ors6':[], 'ors7':[], 'ucct':[], 'Ucct':[],
 'or15':[], 'or17':[], 'or19':[], 'or20':[], 'or21':[], 'or22':[],
 'date':[], 'dat2':[], 'dat6':[], 'wkdy':[], 'bshr':[], 'hldy':[],
 'wkds':[], 'wkdo':[], 'mnts':[], 'mnto':[],
 'yea2':[], 'mnt2':[], 'mnt6':[], 'day2':[], 'day5':[],
 'hrs2':[], 'hrs4':[], 'min2':[], 'min4':[], 'scn2':[], 'DPrt':[],
 'DPnb':[], 'DPmm':[], 'DPbn':[], 'DPod':[], 'DP10':[], 'DPoh':[],
 'excl':[], 'exc2':[], 'exc3':[], 'exc4':[], 'exc5':[], 'exc6':[],
 'null':[], 'copy':[], 'shfl':[], 'eval':[], 'ptfm':[]},
 assignparam = {'default_assignparam' :
            {'(category)' : {'(parameter)' : 42}},
            '(category)' : {'(column)' : {'(parameter)' : 42}}},
 assigninfill = {'stdrdinfill':[], 'MLinfill':[],
            'zeroinfill':[], 'oneinfill':[],
            'adjinfill':[], 'meaninfill':[], 'medianinfill':[],
            'modeinfill':[], 'lcinfill':[], 'naninfill':[]},
 assignnan = {'categories':{}, 'columns':{}, 'global':[]},
 transformdict={}, processdict={}, evalcat=False,
 privacy_encode = False, printstatus=True)
```

Or for postmunge(.) with full range of parameters:

```
test, testID, testlabels, \
labelsencoding_dict, postreports_dict = \
am.postmunge(postprocess_dict, df_test,
 testID_column = False, labelscolumn = False,
 pandasoutput = False, printstatus = True,
 TrainLabelFreqLevel = False, featureeval = False,
 driftreport = False,
 LabelSmoothing = False, LSfit = False, inversion = False,
 traindata = False,
 returnedsets = True, shuffletrain = False)
```

# B   ASSIGNING TRANSFORMS AND INFILL

Assigning root categories is conducted in assigncat parameter and assigning infill in assigninfill - e.g. for a train set df_train with column headers 'col1' and 'col2' we could assign retain normalization (retn) and an integer encoding set (ntgr) with infill types zero infill and ML infill.

Note any columns we don't explicitly assign will defer to automation or we could turn off automated defaults for pass-through of other columns by passing automunge parameter powertransform = 'excl'.

```
assigncat = {'retn':['col1'], 'ntgr':['col2']}
assigninfill = {'zeroinfill':['col1'], 'MLinfill':['col2']}

train, trainID, labels, \
validation1, validationID1, validationlabels1, \
validation2, validationID2, validationlabels2, \
test, testID, testlabels,\
labelsencoding_dict, finalcolumns_train, finalcolumns_test,\
featureimportance, postprocess_dict \
= am.automunge(df_train,
  assigncat = assigncat,
  assigninfill = assigninfill,
  pandasoutput = True)
```

## C  CUSTOM FAMILY TREES

Custom defined family trees of transformations can be passed to a function call by way of the transformdict and processdict parameters. The transformdict parameter is used to populate a family tree, and the processdict parameter is to populate a supporting data structure for a new transformation categories.

Let's demonstrate a scenario to assemble a transformation set in which a normalization is supplemented by two types of bin aggregation. We'll create a new root category 'newt' and populate with transformations pre-defined in the library. Here we'll apply an upstream retain normalization and power of ten bins, and a standard deviation bins downstream of the retain normalization. We'll also include a NArw transformation which designates markers for entries that were subject to infill based on values of the source column.

```
transformdict = {'newt' : {'parents' : ['newt'],
                           'siblings' : [],
                           'auntsuncles' : ['pwr2'],
                           'cousins' : ['NArw'],
                           'children' : [],
                           'niecesnephews' : [],
                           'coworkers' : [],
                           'friends' : ['bins']}}
```

The corresponding processdict will make use of transformation functions defined in the library for the 'retn' (retain) normalization. Here NArowtype designates the types of entries from the source column that will be targets for infill, MLinfilltype designates the types of predictive models to be trained for ML infill, and labelctgy is a support entry for feature importance for cases where a label in returned in multiple configurations.

```
processdict = {'newt' : {'functionpointer' : 'retn',
                         'NArowtype' : 'numeric',
                         'MLinfilltype' : 'numeric',
                         'labelctgy' : 'newt'}}
```

We can then pass these populated structures to a function call and assign a column with header 'col1' to the newly defined root category 'newt'. If we want we can also apply ML infill even on this custom defined transformation set.

```
train, trainID, labels, \
validation1, validationID1, validationlabels1, \
validation2, validationID2, validationlabels2, \
test, testID, testlabels, \
labelsencoding_dict, finalcolumns_train, finalcolumns_test, \
featureimportance, postprocess_dict \
= am.automunge(df_train,
  assigncat = {'newt':['col1']},
  assigninfill = {'MLinfill':['col1']},
  transformdict = transformdict, processdict = processdict,
  pandasoutput=True)
```

When it comes time to process additional data, all of these customizations will be saved in the returned postprocess_dict, which we can then pass to the postmunge(.) function for consistent processing of a test data set.

```
test, testID, testlabels, \
labelsencoding_dict, postreports_dict = \
am.postmunge(postprocess_dict, df_test, pandasoutput=True)
```

# D   DATA AUGMENTATION

The recommended workflow for applying training data augmentation is to assign any desired target columns to DP transforms in assigncat in an automunge(.) call, passing any parameters to assignparam to deviate on default noise distributions if desired, and then process the same set again in postmunge(.) without noise injection, concatenating the two results.

For the example of a received train set df_train with column headers of numeric sets 'number1', 'number2' and column headers of categoric sets 'categoric1', 'categoric2', we can elect to apply noise injections to these columns by assigning DP transforms in assigncat.

```
assigncat = \
{'DPrt':['number1', 'number2'],
 'DP10':['categoric1', 'categoric2']}
```

The noise injection transforms have default entries noted in READ ME for noise ratios and distribution parameters. If we want to overwrite for the transformation category globally can apply assignparam with a default_assignparam entry.

```
assignparam = \
{'default_assignparam' :
    {'DPrt' : {'sigma' : 0.05, 'flip_prob' : 0.5},
     'DP10' : {'flip_prob' : 0.5}
    }
}
```

Or we can overwrite for a specific column, here we demonstrate applying scaled Laplace distributed noise instead of Gaussian to the 'DPrt' transform application to column 'number2'.

```
assignparam.update(
  {'DPrt' : {'number2' : {'noisedistribution' : 'laplace'}}}
)
```

We can then process our train set in automunge(.) with noise injection. Here passing the parameter labels_column = True signals that the final column is to be returned in the labels set.

```
train, trainID, labels, \
validation1, validationID1, validationlabels1, \
validation2, validationID2, validationlabels2, \
test, testID, testlabels, \
labelsencoding_dict, finalcolumns_train, finalcolumns_test, \
featureimportance, postprocess_dict \
= am.automunge(df_train,
               labels_column = True,
               assigncat = assigncat,
               assignparam = assignparam,
               pandasoutput = True,
               printstatus = False)
```

The convention for noise injection transforms in the DP family is that noise is injected to train sets by default and not to test sets unless signaled by the traindata parameter in postmunge(.). Test sets passed to automunge(.) do not receive injections.

Once the training data set has been processed with noise, the same data set can be processed again in the postmunge(.) function and concatenated, using the postprocess_dict dictionary returned from the corresponding automunge(.) call. The postmunge accepts a parameter traindata which signals to the DP whether to inject noise in postmunge(.). traindata is a Boolean defaulting to False for no noise injection. When set as True postmunge(.) will inject noise. Note that each application will inject a unique randomization of noise that is independent of the randomseed parameter although consistent with any distribution parameters.

```
test, testID, testlabels, \
labelsencoding_dict, postreports_dict = \
am.postmunge(postprocess_dict, df_train,
             pandasoutput = True,
             traindata = False,
             printstatus = False)
```

We can then concatenate the results for the train data, ID sets, and label sets to double the number of samples to include a set with and without noise.

```
train = pd.concat([train, test], axis=0, ignore_index=True)
trainID = pd.concat([trainID, testID], axis=0, ignore_index=True)
labels = pd.concat([labels,testlabels], axis=0, ignore_index=True)
```

Although in regular use the automunge(.) function is intended as a resource to apply a train test split based on the valpercent1 and valpercent2 parameters, with data augmentation we recommend extracting validation entries prior to passing a df_train so that the same df_train can be redundantly encoded with or without noise in adjacent calls. The extracted validation set can be processed separately in postmunge(.) or can be passed as a df_test to automunge(.).

Note that while our experiments only doubled the number of samples to include a set with and without noise injection, further multiples of noise injected samples may also be applied by processing in postmunge(.) with traindata=True. The noise distribution properties will be consistent with any parameters passed in the corresponding automunge(.) call. In our experiments we found that too much redundancy had potential to interfere with training convergence.

In an alternate workflow, when only a doubling of the number of samples is desired, the processing can all be performed in a single automunge(.) call by passing the same df_train set as both the train and test sets as follows:

```
train, trainID, labels, \
validation1, validationID1, validationlabels1, \
validation2, validationID2, validationlabels2, \
test, testID, testlabels, \
labelsencoding_dict, finalcolumns_train, finalcolumns_test, \
featureimportance, postprocess_dict \
= am.automunge(df_train,
               df_test = df_train,
               labels_column = True,
               assigncat = assigncat,
               assignparam = assignparam,
               pandasoutput = True,
               printstatus = False)

train = pd.concat([train, test], axis=0, ignore_index=True)
trainID = pd.concat([trainID, testID], axis=0, ignore_index=True)
labels = pd.concat([labels,testlabels], axis=0, ignore_index=True)
```

# E    COLUMN TYPES OF RETURNED DATA

The data returned from an automunge(.) call is intended to be suitable for direct application of machine learning in the framework of a user's choice. In some cases, downstream machine learning libraries may accept as input designations for the types of data found in a column, such as for instance if a column contains a numeric or categoric set. These type of designations may be used for determination of whether to apply an entity embedding layer for instance.

The automunge(.) function thus returns a report of data types for returned columns, available in the returned dictionary as postprocess_dict['columntype_report'].

The report lists column headers of the retuned columns, aggregated by different types of column contents. Specifically, it lists column headers for column content types:

- continuous
- boolean
- ordinal
- onehot
- onehot_sets
- binary
- binary_sets
- passthrough

Here the onehot aggregation contains all one-hot encoded columns, while the onehot_sets sub-aggregates those same columns for those that originated from the same transformation, and similarly with the binary and binary_sets aggregations.

## F  INVERSION

The Automunge library includes an inversion option to recover the form of data as prior to transformations. This type of operation might be useful for instance in converting ML predictions back to the original form of labels, or otherwise for recovering data to invert transformations. The inversion operation is performed in the postmunge(.) function by activating the inversion parameter. The method relies on data properties stored in the postprocess_dict which was returned from the corresponding automunge(.) call. For cases where a source column is included in multiple configurations, the inversion operation relies on a heuristic of selecting the shortest path of transformations with full information retention.

As an example of an inversion operation, if we want to recover the form of labels after a ML prediction process, we can pass those predictions as a dataframe or array as:

```
df_invert, recovered_list, inversion_info_dict = \
am.postmunge(postprocess_dict, predictions, inversion='labels',
        LabelSmoothing=False, pandasoutput=True, printstatus=True)
```

Similarly, if we want to recover the form of a training or test dataset for transformation inversions, we can pass as:

```
df_invert, recovered_list, inversion_info_dict = \
am.postmunge(postprocess_dict, train, inversion='test',
        pandasoutput=True, printstatus=True)
```

# G    EXPERIMENT DETAILS

Further details on the experiments are provided to demonstrate statistical significance. The Tables 8 and 9 reported averages of performance metrics with comparison to the raw data scenario. Here additional detail is provided including standard deviation of the metrics and sampled repetition count serving as the basis. We suspect a large contributor to the variance originates from the stochastic sampling of data set partitions in the experiments, as would explain the increasing standard deviations with smaller training set size.

Table 11: Higgs Experiment Details (AUC Average // Standard Deviation // Repetitions)

|  | Raw Data | Z-Score | Retain | Retain with Bins | Retain w/ Noise full | Retain w/ Noise partial | Retain w/ Augment |
|---|---|---|---|---|---|---|---|
| **full data** | 0.8670 | 0.8668 | 0.8667 | 0.8660 | 0.8268 | 0.8624 | **0.8667** |
| St Dev | 0.0002 | 0.0004 | 0.0005 | 0.0013 | 0.0007 | 0.0007 | 0.0015 |
| Repetitions | 6 | 6 | 6 | 6 | 5 | 5 | 6 |
| **5% data** | 0.8436 | 0.8432 | 0.8431 | 0.8433 | 0.7679 | 0.8388 | **0.8453** |
| St Dev | 0.0011 | 0.0011 | 0.0016 | 0.0014 | 0.0016 | 0.0011 | 0.0006 |
| Repetitions | 30 | 30 | 30 | 30 | 30 | 30 | 30 |
| **0.25% data** | 0.7715 | 0.7716 | 0.7716 | 0.7718 | 0.7052 | 0.7668 | **0.7821** |
| St Dev | 0.0024 | 0.0025 | 0.0026 | 0.0029 | 0.0034 | 0.0025 | 0.0020 |
| Repetitions | 100 | 100 | 100 | 100 | 100 | 100 | 100 |

Table 12: Higgs SVC Experiment Details (Accuracy Average // Standard Deviation // Repetitions)

|  | Raw Data | Z-Score | Retain | Retain with Bins | Retain w/ Noise full | Retain w/ Noise partial | Retain w/ Augment partial |
|---|---|---|---|---|---|---|---|
| **Accuracy** | 0.6410 | 0.6410 | 0.6410 | **0.6821** | 0.6201 | 0.6393 | 0.6402 |
| St Dev | 0.0003 | 0.0003 | 0.0003 | 0.0005 | 0.0006 | 0.0003 | 0.0003 |
| Repetitions | 6 | 6 | 6 | 6 | 6 | 6 | 6 |

# H    A FEW HELPFUL HINTS

A few highlights that might make things easier for first-timers:

1) automunge(.) returns sets as numpy arrays by default (for universal compatibility with ML platforms). A user can instead receive the returned sets as pandas dataframes by passing the parameter pandasoutput = True

2) Even if the sets are returned as numpy arrays, you can still inspect the returned column headers with the returned list we demonstrate as finalcolumns_train

3) Printouts are turned on by default, they can be turned off with printstatus=False

4) Note for data sets with just a few rows, such as those demonstrated here, there is a PCA heuristic to apply dimensionality reduction when the number of features is more than 50% of the number of observations in the train set (this is a somewhat arbitrary heuristic). This can be turned off with ML_cmnd = {'PCA_type':'off'}.

5) Speaking of PCA, if you do want to apply PCA, a useful option allows you to exclude from dimensionality reduction boolean or ordinal encoded columns, available with ML_cmnd = {'PCA_cmnd':{'bool_ordl_PCAexcl':True}}.

6) Note that data shuffling is on by default for the train set and off by default for the test sets returned from automunge(.) and postmunge(.). If you want to shuffle the test data in automunge too you can pass shuffletrain = 'traintest'. Or to shuffle the test data returned from postmunge you can pass the postmunge parameter shuffletrain = True.

7) The automated feature importance evaluation is easy to use, you just need to be sure to designate a label column with labels_column = 'column_header'. Then just pass featureselection = True and printouts will return results as well as the returned report featureimportance

8) To ensure that you can later prepare additional data for inference, please be sure to save the returned postprocess_dict such as with pickle library.

9) Importantly, when you pass a train set to automunge(.) please designate any included labels column with labels_column = (label column header string), which may be an integer index for numpy arrays. When you go to process additional data in postmunge, the columns must have consistent headers as those originally passed to automunge. Or if you originally passed numpy arrays, just be sure that the columns are in the right order. If you're passing postmunge(.) data that includes a column originally designated as a label to automunge, just apply labelscolumn = True.

10) Speaking of numpy arrays, if you pass numpy arrays instead of pandas dataframes to the function, all of the column assignments will accept integers for the column number.

11) When applying ML infill, which is based on Scikit-Learn Random Forest implementations, a useful ML_cmnd if you don't mind a little more training time is to increase the number of estimators as e.g. ML_cmnd = {'MLinfill_cmnd':{'RandomForestClassifier':{'n_estimators':1000}, 'RandomForestRegressor':{'n_estimators':1000}}}

12) Note that any columns you want to exclude from processing, you can either assign them to root category 'excl' in assigncat if you don't mind their retention (noting that they will still be included in ML infill so will need numerical encoding), or you can carve them out from the set to be returned in ID sets consistently shuffled and partitioned such as between training and validation data, just pass a list of column headers to trainID_column and/or testID_column. You can also turn off the automated transforms and only perform those designated in assigncat by passing powertransform='excl'

# I    INTELLECTUAL PROPERTY DISCLAIMER

