# OpenReview forum: "Numeric Encoding Options with Automunge"
_ICLR.cc/2021/Conference — Reject_

### Official Review · AnonReviewer3 · 2020-10-15
**Experiments do not convince**

**Rating:** 2
**Confidence:** 4

**Review:**

This paper introduces a number of data preprocessing options for numeric features provided by an open source library automunge. The most of the paper focuses on explaining the specific transformations offered under each option, including normalization, binning, and noise injection. For normalization, a new transformation 'retain' is offered in addition to traditional z-score, min-max etc. The paper uses one section between normalization and binning to explain a notion 'family tree primitives' which is used in the composition of multiple transformations. In experiments, the paper uses the higgs dataset. Three settings of the dataset are used: full data, 5% data and 0.25% data. 6 settings of transformations are used: raw data, z-score, retain, retain with bins, retain with noise injection, and retain with partial noise injection. When averaged over the three settings of the dataset, using raw data leads to suboptimal auc score compared with the other five. It is acknowledged that "the metrics for normalized data were slightly better than raw on average, it is not clear if they were sufficiently statistically significant to draw firm conclusions. " The paper concludes that "any consideration around benefits of feature engineering should distinguish
first by scale of data available for training. When approaching big data scale input with infinite
computational resources there may be less of a case to be made for much beyond basic normalized
input. "

Pros: The code is open sourced and a larger number of data preprocessing options are offered. The library is released as a python package which is easy to access. In the experiments there are certain cases where the preprocessing helps.

Cons:

1. The experiment results do not convincingly demonstrate the utility of the proposed library. First, when using full data of higgs, the auc is highest when using the raw data. It is even better than using z-score or retain to normalize the data. That is contradictory with the suggestion to consider even basic normalization. Second, when small samples of the data are used, the best auc score among all studied transformations is much lower than the auc score on full raw data. That means even with the transformations, the accuracy does not reach on par with simply using full raw data. Then it raises the question whether the setting of using small sample of higgs data matters. Third, even if we do care about those settings, the best performing transformation is z-score only. That is not a contribution of this paper. To conclude, it is not surprising that one can find some scenarios where some transformation improves the accuracy, but the paper has not presented a convincing scenario where the proposed library is useful.

2. The explanation of 'family tree primitives' confuses me. Table 3 does not help as it is full of unexplained terms. Without examples and context it is difficult to know what 'parents' 'siblings' etc. refer to.

3. It remains a question how a user should select transformations from the numerous offered options.

---

> ### Author Response · Authors · 2020-11-12
> **response to reviewer 3**
>
> Thank you reviewer for your assessment of our work. I appreciate your recognition of the value of an open source library with a large number of data preprocessing options, and that you found the library easy to access.
>
> I will briefly here respond to those points that you listed as cons in your review. I hope that given the revisions that we have incorporated into the paper based on feedback from other reviewers as well as material findings from further experiments run since submitting the paper (which I will shortly discuss), you might be willing to reconsider your rating.
>
> 1)
> You offered that the experiments do not convincingly demonstrate the utility of the proposed library. I agree with you assessment based on the results included in the original submission, other than the exception that we demonstrated a material benefit to linear kernel learning associated with bin aggregators in Table 9.
>
> However, since submitting this paper, I’ve had the time to continue running experiments, particularly with respect to noise injection as a source of data set augmentation for improved accuracy in circumstance of limited availability of training data. The revised paper that I have uploaded includes a new column in the experiment results for Tables 8 & 9, and I believe the realized benefit of data set augmentation through the noise injection transformations is a significant and important finding that may be of broad interest to the machine learning community. Our findings indicate that particularly in scenarios of data availability constraints, augmenting the training set with noise injected data materially improves the AUC metric results. Here we show for the 0.25% data scenario that the AUC metric improved from 0.7662 to 0.7890 (+2.9%), for the 5% data scenario improved from 0.8416 to 0.8522 (+1.3%), and the full data scenario only improved marginally from 0.8664 to 0.8672 (+0.09%). I am including a revised paper and a demonstration Jupiter notebook in the new upload.
>
> Although data set augmentation is common for CNN / image applications, I believe this is the first instance of a generalized and automated solution to perform data set augmentation for tabular data sets. I believe this is an important finding which we have experimentally validated, and is available in a downloadable open source library. We expect the machine learning research community would be very interested in this finding.
>
> 2)
> You found the description of family tree primitives confusing. In the new version uploaded with these comments we have added and rewritten some of the paragraphs in section 5 which give concrete description of how the family tree primitives are applied in context of the examples in Figures 1 & 2. I believe these added paragraphs should go a long way for your understanding of the family tree primitives, which we believe is a significant innovation for the Automunge library, one way to think about them is that they are a fundamental reframing of command line specification for multi-transform sets, as may include generations and branches of derivations, which are applied by way of recursion.
>
> 3)
> You suggested that it remains a question how a user should select transformations from the numerous offered options. I’ll note here a few cases where I believe this was addressed, and also have incorporated a new Table 10 into the discussions in closing section 9 which may be a useful summary for this purpose. I think this table was a good addition thank you for this feedback.
>
> For the cases of normalizations discussed in Section 2, we noted in the paper that scaling with (max - min) in the denominator helps when a fixed range of values are desired for the returned set. We noted that retain normalization is of benefit when interpretability of the scaled data is desirable.
>
> For the bins and grainings described in section 4, one of the findings of our experiments was a pronounced benefit to a support vector classifier associated with supplementing normalized numerical sets with aggregated bins, as demonstrated in Table 9.
>
> For the noise injection transforms described in Section 5, we noted repeatedly that noise injections may serve purposes of noise injection for differential privacy, model perturbation for ensemble aggregations, and particularly, as now validated with the new experiment results, may serve the purpose of data set augmentation for cases with underserved training data.
>
> For the sequential data transforms described in section 6, we noted that these are useful for time-series data as may benefit from supplementing numeric streams by proxies for derivatives.
>
> For the integer sets described in section 7, we noted that these are useful for integer sets with unknown interpretations.
>
> I hope that you might consider giving the paper another review. I believe with the revisions to Section 5 to clarify the family tree primitives, the new experiment findings in Table 8, and the new closing Table 10 that this is a much better paper.

---

### Official Review · AnonReviewer2 · 2020-10-27
**Concerns with clarity, novelty, scope of experiments**

**Rating:** 3
**Confidence:** 4

**Review:**

This paper presents a set of numeric normalization features from the Automunge open-source python library, and evaluates them in two machine learners (one neural, one SVM) on a data set.  In general the paper discusses standard normalizers rather than introducing novel techniques, and the experiments are too limited to evaluate the effectiveness of the approaches.

I did not understand the family tree primitives, unfortunately.  I think I need a concrete example.  What do upstream and downstream mean in this context?  What is a “generation”?  Why are the primitive names (the familial relations) appropriate to describe each row in Table 3?

The experiments also do not show that the transformations described in the paper offer benefits for neural models.  In fact, on the one data set and one neural architecture evaluated, using the best normalizations results in essentially the same performance as using raw unnormalized data.  Even if the experimental results had been positive, the limited scope of the investigation would make it impossible to draw general conclusions.  A substantially expanded set of experiments that looked at more neural architectures and many more data sets might give practitioners more useful guidance, especially if it led to concrete recommendations about which normalizations to try for which type of data.

Minor:

classical “or quantum” computation may require non-negative feature values?  Why mention quantum here?

Tables 8 -> Table 8

The author refers to Automunge as “our” library at one point, breaking anonymity.

---

> ### Author Response · Authors · 2020-11-12
> **response to reviewer 2**
>
> Hello reviewer, thank you for your comments. I’ll quickly respond to your primary points of feedback.
>
> The family tree primitives we believe are a particularly useful fundamental reframing of command line specification for sets of a transformations, such as sets that may include generations and branches of derivations, which are implemented by way of recursion in the library. We are uploading a revision to our paper which includes some new paragraphs  and revisions in Section 5 to clarify the application of family tree primitives as demonstrated in the figures 1-4. The table 3 primitives were included to clarify their demonstrations in the figures 1-4 which illustrate the specification of transformation sets associated with the various options for transformation functions present in the library.
>
> Regarding the experiments, I agree that other than the support vector demonstration, no significant finding was demonstrated in these experiments. However, since submitting this paper, I’ve had the time to continue running experiments, particularly with respect to noise injection as a source of data set augmentation for improved accuracy in circumstance of limited availability of training data. The revised paper that I have uploaded includes a new column in the experiment results for Tables 8 & 9, and I believe the realized benefit of data set augmentation through the noise injection transformations is a significant and important finding that may be of broad interest to the machine learning community. Our findings indicate that particularly in scenarios of data availability constraints, augmenting the training set with noise injected data materially improves the AUC metric results. Here we show for the 0.25% data scenario that the AUC metric improved from 0.7662 to 0.7890 (+2.9%), for the 5% data scenario improved from 0.8416 to 0.8522 (+1.3%), and the full data scenario only improved marginally from 0.8664 to 0.8672 (+0.09%). I am including a revised paper and a demonstration Jupiter notebook in the new upload.
>
> Although data set augmentation is common for CNN / image applications, I believe this is the first instance of a generalized and automated solution to perform data set augmentation for tabular data sets. I believe this is an important finding which we have experimentally validated, and is available in a downloadable open source library. We expect the machine learning research community would be very interested in this finding.
>
> Regarding the mention of quantum computing, one of the goals of this paper is to attract users to our library, we welcome users from quantum computing field as well. We believe our comment is accurate.
>
> I corrected the typo you noted thank you for identifying.
>
> I hope that you might consider giving the paper another review. I believe with the revisions to Section 5 to clarify the family tree primitives, the new experiment findings in Table 8, and the new closing Table 10 that this is a much better paper. Best regards.

---

### Official Review · AnonReviewer4 · 2020-10-28
**A new library for tabular data pre-processing for machine learning tasks**

**Rating:** 3
**Confidence:** 4

**Review:**


Summary:

The paper describes a library (Automunge) for pre-processing tabular data to prepare the data for downstream machine learning tasks. The paper also describes how to use the said library and the various options (including some new forms of normalization) available in the library. Experimental evaluation on Higgs Boson interaction dataset is provided, following the experimentation in Baldi, Sadowski and Whiteson 2014. Some improvements over the published results in Baldi 2014 are shown.

Key strengths:

The paper is well written for the most part, apart from a few typos. The appendix provides examples of usage of the proposed library. The supplementary material contained the code and an extensive readme for the code.

Suggestions for improvements:

1. The primary novelty of the library is to ease the burden of tabular data pre-processing. However, it seems that the user has to specify the configuration for the pre-processing unless the default options are acceptable. For most machine learning tasks, the choice of pre-processing can be heavily influenced by the context and the domain expertise. It is unclear how a library can automatically derive these options correctly, without the knowledge of the downstream machine learning task (or indeed the domain).
2. The paper provides evaluation on one dataset (Higgs Boson interactions) to support the claim that the library is of value for many machine learning tasks. The setup of the experiment is slightly different from the original paper (Baldi 2014): i.e. learning rate schedule is different, but network setup is the same. Looking at the results in the original paper (Table 3 of Supp material in Baldi 2014), the results presented in this paper are in the same range as the original paper (but also within the variance of results in the original paper), so it's difficult to extrapolate how significant the improvement due to retain normalization (this paper) is, and how it might generalize to other machine learning tasks, or even other datasets. Also, some results improve upon the original paper, whereas others don't, which makes it harder to say how statistically significant the effect of the pre-processing presented in this paper is.
3. Retain normalization: The paper claims that retain normalization is better because it retains the sign of the original datapoints, but it's unclear why this is important if it doesn't affect the eventual result. Perhaps this could be clarified?
4. Software review: The entire library is written in a single Python file. The naming convention is not Pythonic. From the usage examples and reviewing the code, it's clear that the library makes little use of encapsulation of data structures, leading to unwieldy usage. All documentation is provided as a single Readme, rather than as Sphinx documentation or API documentation. There don't appear to be any tests or continuous integration included. The code is not modularised. There are no docstrings with param/return descriptions or type annotations.

Overall comments:

For what is mainly a description of an automated data pre-processing library, the usage of the library is not sufficiently automatic (i.e. setting up the options amounts to feature engineering, which the paper claims to circumvent). The new normalization technique presented in the paper does not yet demonstrate a clear improvement over current options.

---

> ### Author Response · Authors · 2020-11-12
> **response to reviewer4 (part 1 of 2)**
>
> Hello reviewer, thank you for your suggestions. I will provide here a response.
>
> First, please note that some additional paragraphs are now incorporated into Section 5 to clarify the use of family tree primitives per feedback from reviewer 1. I will also detail at the conclusion of this response some updates based on further investigations since the original paper was submitted, in which we experimentally validate the use of noise injections for data set augmentation for material improvement to model performance, a finding that we expect the machine learning research community will be very interested in as we do not believe a generalized automated solution yet exists for tabular data augmentation, as is now available in our open source library.
>
> Regarding your suggestions for improvements:
>
> 1)
>
> The library is intended as a resource for a range of use-cases including full automation, specification of transformations from our library, as well as a platform for custom defined transformation functions. It very easily mixes and matches between these scenarios. Under full automation, the library simply performs normalization of numerical sets, binary encoding of categoric, and extraction and encoding of time scales for date-time data. For numerical data as is the focus of this paper, under full automation a user can defer to z-score normalization, activate a parameter for universally supplementing z-score normalized data with aggregated standard deviation bins (as is showed to be used for kernel methods in Table 9), or an additional option not discussed in the paper is to activate a parameter to evaluate distribution properties of numeric sets to select between normalization options such as z-score normalization, min-max scaling, and for high skew data a box-cox power law transformation.
>
> We agree with your assessment that the choice of pre-processing can be heavily influenced by the context and the domain expertise. There are some libraries, such as the fastai library that we applied in our experiments, where no options are provided for numeric preprocessing other than basic scaling. The whole point of this paper is to survey the various options available for users who wish to pursue a more fine-grained evaluation of preprocessing methods associated with each feature set as may be domain dependent. The Automunge library supports very simple methods for specifying transformations and/or transformation sets. Assigning transformations is demonstrated in Appendix B and composing custom transformation sets is demonstrated in Appendix C.
>
> Whats more, the transformation functions as may be included in a set need not be defined internal to the library. We have documented in our READ ME a convention for defining custom transformations functions that may be externally defined and incorporated directly into an automunge(.) call, such as may then take advantage of all of the useful methods such as automated autoML derived missing data infill, automated dimensionality reductions and transformations, and pushbutton data pipeline operation for consistently processing subsequent data.
>
> The choice between full automation, specification from our library, or custom defined transformations may be mixed and matched for the various features found within a data set. The family tree primitives discussed in the paper are particularly useful for specifying sets of transformations as may include generations and branches of derivations.
>
> 2)
> For the most part I agree with your description of the experimental findings. Some clarifications I would like to call out are that for the support vector with linear kernel scenario demonstrated in Table 9 the aggregation of standard deviation bins was shown to have a material improvement on performance - I suspect there may remain some domains where linear kernels may still be of interest to researchers.
>
> I noted at the start of this response that since submitting the paper we have continued our experiments, and I am pleased to share that we have now experimentally validated a material improvement to model performance by way of noise injection as a source of data set augmentation. Please refer to the additional column added to Table 8 in the new draft. We believe that the solution of an automated source of tabular dats set augmentation is a novel invention and we expect that the machine learning research community will be interested in this finding.

---

> > ### Author Response · Authors · 2020-11-12
> > **response to reviewer4 (part 2 of 2)**
> >
> > 3)
> > Regarding your assessment of retain normalization, the paper noted that model performance is not the only benefit, and that interpretability of returned data is also worthy of consideration. Consider our use of the retain normalization for the dxdt family of transforms described in Section 6 on Sequential Data. We have demonstrated means to extract proxies for first, second, and higher order derivatives associated with numeric sets in a time-series application. The output of each of the intermediate steps in our demonstration is subject to a retain normalization, and we believe the usefulness of the sign retention for interpretability is particularly of benefit for this use case.
> >  4)
> > Regarding your software review, you noted that the library is written in a single python file, without a pythonic naming convention, with little use of data structure encapsulation, and noted lack of tests or continuous integration, as well as a few other points.
> >
> > I greatly appreciate this type of feedback, this is part of the value of submitting work to a research conference to receive this type of input, which I am taking to heart as potential areas for improvement on the library. I would like to call out that some of these considerations that you noted were actually somewhat intentional, from the standpoint of development environment. By encapsulating our code-base in a single file we are able to approach development in a live-coding environment, for which we use a hybrid of VSCode for editing the master library and Jupyter notebook for running validations and evaluating outputs associate with any changes. This type of live-coding environment we have found particularly useful for rapidly iterating. The Jupyter notebook aspect of our development environment in particular has a (nearly) comprehensive set of validations and tests that are performed prior to each software update, and at least for now while we are still rapidly iterating this is our preferred development environment. The READ ME documentation we consider thoroughly vetted and updated corresponding to each new rollout, it is quite comprehensive and includes hyperlinks to help navigate.
> >
> > Regarding your overall comments, I believe you may have missed a few of our contributions, as you seem to credit the chief contribution to be the retain normalization. I’ll list here a few of the improvements to mainstream practice demonstrated in this paper:
> > - The noise injection options we believe to be a significant contribution, especially for purposes of data set augmentation which we now have experimentally validated with the new column in the experiment results of Table 8, discussed further below.
> > - The dxdt family of transforms discussed in section 6 are a particularly useful option to supplement time-series evaluations with proxies for derivatives of numeric streams.
> > - The Automunge library in general is an entire reframing of the workflow associated with preprocessing tabular data, we believe the family tree primitives as demonstrated throughout this paper are a significant innovation for command line specification of transformation sets via recursion.
> >
> >
> > Finally, as I eluded to earlier in my response, since submitting this paper, I’ve had the time to continue running experiments, particularly with respect to noise injection as a source of data set augmentation for improved accuracy in circumstance of limited availability of training data. The revised paper that I have uploaded includes a new column in the experiment results for Tables 8 & 9, and I believe the realized benefit of data set augmentation through the noise injection transformations is a significant and important finding that may be of broad interest to the machine learning community. Our findings indicate that particularly in scenarios of data availability constraints, augmenting the training set with noise injected data materially improves the AUC metric results. Here we show for the 0.25% data scenario that the AUC metric improved from 0.7662 to 0.7890 (+2.9%), for the 5% data scenario improved from 0.8416 to 0.8522 (+1.3%), and the full data scenario only improved marginally from 0.8664 to 0.8672 (+0.09%). I am including a revised paper and a demonstration Jupiter notebook in the new upload.
> >
> > Although data set augmentation is common for CNN / image applications, I believe this is the first instance of a generalized and automated solution to perform data set augmentation for tabular data sets. I believe this is an important finding which we have experimentally validated, and is available in a downloadable open source library. We expect the machine learning research community would be very interested in this finding.
> >
> > I hope that you might consider giving the paper another review. I believe with the revisions to Section 5 to clarify the family tree primitives, the new experiment findings in Table 8, and the new closing Table 10 that this is a much better paper.

---

### Official Review · AnonReviewer1 · 2020-11-01
**This work may be interesting, but the paper doesn't show why**

**Rating:** 2
**Confidence:** 4

**Review:**

This paper reviews an existing Python feature engineering library, Automunge. It describes Automunge's functions operating on numeric (as opposed to categorical) input data. It describes notions of transformations, operations available to bin, inject noise, process sequential data and integer sets.

The Automunge library implements what the data scientist might expect from a library of feature transforms for numerical data. It is not particularly rich or original. The library structure is not made clear in the paper. It is a pity that the paper does not compare to any other feature engineering library, although there are very many available.

The experiments section reports two experiments on one dataset; it reports AUC and accuracy metrics under different feature pre-processing: this reporting is hardly relevant and does not demonstrate any interesting property of Automunge. For instance, it would be interesting to see whether Automunge is particularly flexible, clear, or convenient when one wants to implement chains of transformations. It is not clear why, table 8, it is interesting to observe 100%, 5% and 0.25% of data regimes, nor why we should average over these three.

Tables 3 and 5, and none of the figures are mentioned in the text, let alone commented upon. The figures are left unexplained and contain undefined abreviations (NArw, DPo3, DPo6). I could not make sense of them. The family tree structure which seems to be the object of Table 3 is not explained, and I could not make sense of it, though it seems to be important.

In table 4, I could not understand what the columns refer to; equally, the transformations mentioned there are left undefined: for instance, in case, as I assume, "Number of standard deviations from the mean" is $x \mapsto round(|(x_i - µ) / \sigma) |)$ (or maybe the absolute value needs to be replaced by round brackets? this is left ambiguous), it should be properly defined. Similarly, what is "Powers of ten"? Does it mean $x \mapsto round(\log_{10} x) $ ? The id strings are cryptic. It is not clear why identifiers seem to be all limited to four characters; this makes the entire naming scheme very difficult to follow, as abreviations seem arbitrary; in addition, this is contrary to Python variable naming conventions, particularly naming conventions in popular and successful ML libraries, which prefer explicit and long-form function and argument naming over abreviated and cryptic ones.
Table 6 refers to categoric noise injections, but the paper was meant to be about numeric input variables, so I'm not clear what it is doing here.

The text is hard to understand. This is in part, but not only, due to long-winded, unclear sentences: for instance the very first sentence is obscure and ill-structured, with several syntax errors, for instance the repeated and often incorrect usage of "such as". Throughout the text, syntax issues make the text hard to understand. Examples of this:
- "data transformations ... are to be directed for application to a distinct feature set as input"
- such as multiplicative properties at, above, or below.
- potentially including custom defined transformation functions with minimal requirements of simple data structures
- the last sentence of sec3
- the first sentence of sec4 is needlessly intricate and seems to say nothing more than "Automunge transformations are invertible".
- the paragraph below figure 2
- a given ratio of input entries are flipped to one of the other encodings between which have a uniform probability

The very last conclusion paragraph is surprising and seems irrelevant. Similarly, the paper mentions quantum computation for reasons that escape my comprehension, section 2.

The paper is so hard to understand that it does not allow one to evaluate possible upsides of Automunge on its own merit. After quite some hesitation, I have to assign this paper a severe rating due to the conjunction of several severe shortcomings.

---

> ### Author Response · Authors · 2020-11-12
> **detailed response**
>
> Thank you reviewer for your extensive review. I hope you forgive the length of this response, you gave a quite detailed assessment of various points, and I will try to address them one by one. As part of this response there were several updates made to the paper, so I hope you will consider taking another look.
>
> As a general comment, the paper attempted to cover a lot of ground, and I think part of your concerns stem from the at times parsimonious treatment of various aspects that arose as a result.
>
> I’m also going to note at the conclusion of this response a particular update of note to the experiments section demonstrating what we consider a significant finding with respect to automated data set augmentation for tabular data by way of the noise injection transforms described in the paper, which we believe may be of wide interest to the machine learning community, and thus may more than offset the various concerns that you raise, I hope that you may take into account this finding in your consideration.
> __
>
> “The Automunge library implements what the data scientist might expect from a library of feature transforms for numerical data. It is not particularly rich or original.”
> => we believe several aspects to the library detailed in the paper are original, as for example
> - Retain normalization,
> - New kinds of “family tree” primitives for specifying multi-transform sets that may include generations and branches of derivations implemented by way of recursion,
> - A single uniform convention for extracting binned aggressions of a numeric sets into various categoric encoding types such as ordinal, one hot, and binary
> - New kinds of sequential transforms to extract a proxy for first, second and higher order derivatives of a numeric time series stream by the dxdt transforms,
> - The noise injection transformations in particular are an important point of originality, as demonstrating an automated way to apply data set augmentation to tabular data, an important finding.
> - Although not discussed in the paper, a particular point of novelty that is worth keeping in mind comes from the manner in which automunge(.) through processing populates and returns a python dictionary “fit” to properties of the train set containing all of the steps and parameters of transformations, that may then be passed to the postmunge(.) function for consistent and efficient processing of subsequent data on the train set basis - we believe the machine learning community would benefit from this new convention for preprocessing tabular data by giving researchers a means to publish and share their processing experiments for reproducibility of benchmarks and experiments.
>
> “The library structure is not made clear in the paper.”
> => The paper was intended as a survey of numeric transforms. Other papers have been published by the team describing the library structure and architecture, there’s a good one I could suggest but it would be difficult to anonymize.
>
> “The experiments section reports two experiments on one dataset; it reports AUC and accuracy metrics under different feature pre-processing: this reporting is hardly relevant and does not demonstrate any interesting property of Automunge.”
> => We included the “raw” scenario as benchmark for normalization in general. We included z-score normalization as a benchmark against retain normalization. We included the supplemental bins scenario because of their pronounced benefit for a linear kernel as demonstrated in Table 9. We included the noise injection (full and partial scenarios) as an informative indication of noise impact to accuracy in comparison to retain normalization. We have added a column to these tables 8 & 9 demonstrating the impact of data set augmentation via noise injection to supplement normalized data, which turns out to have a material benefit to performance which finding we believe may be of wide interest to the machine learning community.
>
> “it would be interesting to see whether Automunge is particularly flexible, clear, or convenient when one wants to implement chains of transformations”
> => This aspect is one of the strengths of the family tree primitives. We have attempted to illustrate the operation for chains of transformations making use of the family tree primitives by way of Figures 1, 2, 3, and 4. As part of our response we have also extensively revised section 5 to clarify their use.
>
> “It is not clear why, table 8, it is interesting to observe 100%, 5% and 0.25% of data regimes, nor why we should average over these three.”
> => This was included as an interesting illustration of impact of underserved data, and turns out to be a useful framing to illustrate the impact of data set augmentation, which appears to have even more pronounced benefit for smaller data set scenarios. The averaging over these three was not a key finding, it was included as a summary, but based on your feedback I have dropped the averaging row for improved clarity, thank you for the suggestion.

---

> > ### Author Response · Authors · 2020-11-12
> > **detailed response part 2**
> >
> >
> > “Tables 3 and 5, and none of the figures are mentioned in the text, let alone commented upon.”
> > => Thank you for pointing this out. Regarding table 3, the family tree primitives are made note of in the preceding paragraph, I’ve added a reference to the table for clarity - their inclusion supports the illustrations of Figs 1-4. Regarding table 5, it is referenced in the next to last paragraph of section 5. Regarding the Figures 1-4, I’ve added reference to them in the text. In general, the figures were meant to illustrate the use of family tree primitives to specify “chains of transformations” as well as to provide further clarification on the transformation compositions described in the text.
> >
> >
> > “The figures are left unexplained and contain undefined abbreviations (NArw, DPo3, DPo6).”
> > => Thank for this feedback. I’ve added a ‘NArw’ column to Figure 1 for clarity. The ‘DPo6’ and ‘DP03’ referenced in Fig2 are defined by way of the note of “(support entries for DP10)”, such that DPo6 is a transformation category entry to the DP10 parents primitive and associated with a ord3 transformation function per the entry in the DP06 processdict which is where the ‘_ord3’ suffix comes about for the returned column ‘column_ord3_DPod_1010’ returned from the DP10 root category. Similarly for DPo3. These are now clarified in the text - I’ve added a paragraph to Section 5 to describe family tree primitive applications for root category DPmm in Figure 1 based on your feedback. I’m hoping this goes a long way to clarifying their application - if you still think this needs further clarification please let me know. I’ve also rewirtten other parts of section 5 to better clarify.
> >
> >
> > “The family tree structure which seems to be the object of Table 3 is not explained, and I could not make sense of it, though it seems to be important.”
> > => Some further clarification for family tree primitives is offered by Fig’s 4,5,6,7 of the cited paper by same author which I included a preprint of in the supplemental material. I’m hoping that the rewrite of section 5 helps here as well.
> >
> >
> > “In table 4, I could not understand what the columns refer to”
> > => The second paragraph of section 4 notes that binning options may return categoric encodings as one-hot, ordinal, and binary encodings. I’ve added a clarification to the first sentence of this paragraph to clarify that the entries shown under those three headers are transformation categories. I’ve also updated the Table 4 header from “Binning Options” to ‘Binning Transformation Category Options”. Thanks for this feedback.
> >
> > “the transformations mentioned there are left undefined”
> > => I was hoping that it was self-evident from these being served as categoric encodings, as well was from the use of terms like “coarse graining” “aggregate buckets” and “binning”, that these are rounded into distinct buckets. The first column defines the transformations and the columns one-hot / ordinal / binary distinguish how the categoric encoding is served for the listed transformation categories.
> >
> >
> > “The id strings are cryptic. It is not clear why identifiers seem to be all limited to four characters; this makes the entire naming scheme very difficult to follow, as abreviations seem arbitrary; in addition, this is contrary to Python variable naming conventions, particularly naming conventions in popular and successful ML libraries, which prefer explicit and long-form function and argument naming over abreviated and cryptic ones.”
> > => One of the benefits of the four character string convention for transformation categories is the ability to include the entire catalog in demonstrations such as that shown in Appendix A. Automunge has an extensive library of transformations well beyond what is discussed in this paper, and the intended resource for help navigating these abbreviations is the library of transformations section in the READ ME uploaded to supplemental material, where transformation categories are aggregated by different conventions and have full descriptions. A further benefit is that these same 4 character strings are appended to the returned column headers logging the steps of transformations for each returned column, such as for chains of transformations. There is nothing architecturally that prevents longer strings to serve this purpose, this was a style choice. Kind of like when Apple just gives you one button on the mouse, there are always tradeoffs.
> >
> > “Table 6 refers to categoric noise injections, but the paper was meant to be about numeric input variables, so I'm not clear what it is doing here. ”
> > => The paper gives several examples of numeric encodings in which binned aggregations are served as categoric encodings in section 4. Also since one of the key findings is the use of noise injections for data set augmentation it was deemed appropriate to include categoric noise injections in the writeup for completeness.

---

> > > ### Author Response · Authors · 2020-11-12
> > > **detailed response part 3**
> > >
> > > “the very first sentence is obscure and ill-structured… for instance the repeated and often incorrect usage of "such as" ”
> > > => I slightly tweaked the first line, hoping it reads better now. Wow I just counted them you are right, my overuse of ‘such as’ is probably a relic from working a lot in legal documents lately - I’ve now removed or replaced a lot of these instances, thanks for suggesting.
> > >
> > > “Throughout the text, syntax issues make the text hard to understand. Examples of this: ”
> > >
> > > "data transformations ... are to be directed for application to a distinct feature set as input" => this reads as intended, hope you don’t mind if I leave this one in place
> > >
> > > “such as multiplicative properties at, above, or below.”
> > > => changed this to “at/above/below” hope it reads better this way.
> > >
> > > “potentially including custom defined transformation functions with minimal requirements of simple data structures”
> > > => I rephrased this as “potentially including custom transformation functions which may be defined with minimal requirements of simple data structures”
> > >
> > > “the last sentence of sec3”
> > > => Thanks for feedback I broke this sentence into two and added the title of the classification for clarity (“NArowtype”).
> > >
> > > “the first sentence of sec4 is needlessly intricate and seems to say nothing more than "Automunge transformations are invertible". ”
> > > => information retention properties of transformation functions is an important distinction and comes into play with some of our data structures, I hope you don’t mind if I leave this clarification in place.
> > >
> > > “the paragraph below figure 2”
> > > => I’m hoping that the additional paragraph added after this one helps clarify the content.
> > >
> > > “a given ratio of input entries are flipped to one of the other encodings between which have a uniform probability”
> > > => I rewrote this paragraph for clarity
> > >
> > > “The very last conclusion paragraph is surprising and seems irrelevant.”
> > > => We believe this is relevant in the context of the findings of Table 9, particularly the 4% improvement to accuracy associated with supplementing numeric sets with standard deviation bins as demonstrated for the SVC with linear kernel.
> > >
> > > “the paper mentions quantum computation for reasons that escape my comprehension, section 2. ”
> > > => sorry if this caught you off guard, I write about quantum computing in a lot of my research. There are some optimization algorithms applied for quantum computing, such as I believe quantum annealing, in which non-negative numeric sets are a prerequisite.
> > >
> > > “The paper is so hard to understand that it does not allow one to evaluate possible upsides of Automunge on its own merit ”
> > > => I hope that the revisions to section 5 may go a long way to clarifying the use of family tree primitives and their value as a novel means to specify transformation sets which may include generations and branches of derivations. We believe this innovation is a particularly useful and fundamental reframing of data processing specification.
> > >
> > > “After quite some hesitation, etc”
> > > => I’ve tried to go through your comments one by one and provide clarifications and revisions per your suggestions, I hope you might take this into account and reconsider this rating. I hope you also might take into account the following:
> > >
> > > Since submitting this paper, I’ve had the time to continue running experiments, particularly with respect to noise injection as a source of data set augmentation for improved accuracy in circumstance of limited availability of training data. The revised paper that I have uploaded includes a new column in the experiment results for Tables 8 & 9, and I believe the realized benefit of data set augmentation through the noise injection transformations is a significant and important finding that may be of broad interest to the machine learning community. Our findings indicate that particularly in scenarios of data availability constraints, augmenting the training set with noise injected data materially improves the AUC metric results. Here we show for the 0.25% data scenario that the AUC metric improved from 0.7662 to 0.7890 (+2.9%), for the 5% data scenario improved from 0.8416 to 0.8522 (+1.3%), and the full data scenario only improved marginally from 0.8664 to 0.8672 (+0.09%). I am including a revised paper and a demonstration Jupiter notebook in the new upload.
> > >
> > > Although data set augmentation is common for CNN / image applications, I believe this is the first instance of a generalized and automated solution to perform data set augmentation for tabular data sets. I believe this is an important finding which we have experimentally validated, and is available in a downloadable open source library. We expect the machine learning research community would be very interested in this finding.
> > >
> > > I hope that you might consider giving the paper another review. I believe with the revisions to Section 5 to clarify the family tree primitives, the new experiment findings in Table 8, and the new closing Table 10 that this is a much better paper.

---

### Author Response · Authors · 2020-11-14
**final rebuttal version uploaded**

Ok this note intended for all reviewers. Thank you again for you feedback I think it has helped to make this paper much stronger.

So to review, I previously noted the following revisions:
- extensive revisions to Section 5 to clarify family tree primitives
- new experiment findings added to the experiment results detailing the novel innovation of generalized data augmentation for tabular learning resulting in improved model performance for cases of underserved training data
- new Table 10 at conclusion summarizing use cases

This final rebuttal version has also incorporated following:
- shortened the title (removed the phrase "A Numbers Game")
- added this sentence to the abstract: "Experiments demonstrate the realization of a novel generalized solution to data augmentation by noise injection for tabular learning, as may materially benefit model performance in applications with underserved training data."
- added final metric averages to full data scenarios with data augmentation for Tables 8 and 9
- made the key findings in Tables 8 and 9 bold text for clarity
- otherwise a few small cleanups and corrections
- also updated the code base in supplemental material from version 4.88 to the current version 5.22

Thank you again. I welcome any further questions or feedback. Best regards.

---

> ### Author Response · Authors · 2020-11-15
> **added Appendix D to demonstrate Data Augmentation**
>
> Hi reviewers, sorry another quick update. Thought it would be beneficial for clarity to add an additional Appendix to formally demonstrate data augmentation workflow. That is now available as the new Appendix D. I also updated the data augmentation demonstration notebook in supplemental material to a simplified version consistent with the appendix. Best regards.

---

> > ### Author Response · Authors · 2020-11-16
> > **Disregard recent upload’s data augmentation experiments**
> >
> > Hi, kind of embarrassing, recent update with reported results for data augmentation in table 8 were a result of data leakage as validation set was extracted after noise injection duplication. Silly mistake. Rerunning the experiments and expect to have initial results ready by Tuesday without data leakage.

---

> > > ### Author Response · Authors · 2020-11-16
> > > **Results still impressive**
> > >
> > > Hi, wanted to share that results after correcting for data leakage are now uploaded for data augmentation. Results are still impressive, just more realistic. Please refer to Table 8 and 9. The results for the full data scenario are pending, to follow. Best regards.

---

> > > > ### Author Response · Authors · 2020-11-17
> > > > **Final final rebuttal version**
> > > >
> > > > Hello reviewers, thanks for your patience as we finalized the data augmentation experiments for the full data scenario. Those results are now uploaded and in line with expectations (data augmentation had negligible effect on full data scenario, increasing benefit to model performance as the amount of training data was reduced).
> > > >
> > > > The only other edit is was a small change to the experiments section replacing a reference to the averages of results, which averages were stricken from table 8 based on one of reviewer's comments, now replaced now with an observation that normalizations appeared to have greater benefit in cases of underserved training data.
> > > >
> > > > Please note that the supplemental material has a demonstration notebook for the data augmentation experiments including the correction for data leakage as file "HIGGs_demonstration_DataAugmentation_111620_fixeddataleakage.ipynb".
> > > >
> > > > Happy to answer any further questions you may have.
> > > >
> > > > Best regards

---

> > > > > ### Author Response · Authors · 2020-11-18
> > > > > **Running additional repetitions to improve statistical significance**
> > > > >
> > > > > Since I believe there is another week before final submissions are due, am now running some additional repetitions on the experiments to try and improve the statistical significance of the reported averages of the performance metrics. Hoping in the next week can increase the number of repetitions on each scenario from 3 to 6. Every little bit helps. Best regards.

---

> > > > > > ### Author Response · Authors · 2020-11-19
> > > > > > **New Appendix detail experiment statistics**
> > > > > >
> > > > > > As an update on progress, I just uploaded a revision capturing following updates:
> > > > > >
> > > > > > 1) small update to the Data Augmentation Appendix E demonstrating the designation of a label column in context of the workflow
> > > > > >
> > > > > > 2) Added a new Appendix G detailing statistics of the experiments, including average, standard deviation, and repetition count. The hope is that by the deadline will have sufficient repetitions included to remove the statement in the Experiments section regarding statistical significance of the results. Currently anticipate that will have sufficient time to include ~6 repetitions of the full data scenario, ~18 repetitions of the 5% data scenario, and ~100 repetitions of the 0.25% data scenario.
> > > > > >
> > > > > > Best regards.

---

### Author Response · Authors · 2020-11-23
**Improved Statistical Significance**

Hello reviewers. Please accept my sincerest gratitude for tolerating my updates through the review period. As may have been apparent, the validation of data augmentation by noise injection has been coalescing in real time through these updates, and I believe it is now well vetted. As noted in a prior comment, I have spent the last week running repetitions on these experiments to improve statistical significance of the findings. As a result there has been some aspects of reversion to the mean for the various scenarios, which I believe is a positive development as now have a more accurate representation for these findings.

Please find attached what we promise is the real final version of the paper, which includes the following revisions:
- Updates to performance metrics reported in Tables 8 and 9 based on experiment repetitions to improve statistical significance
- Corresponding updates to Appendix G further detailing statistics and repetition counts
- Two small edits to Section 8 - Experiments:
    - Removed this sentence: “Although the normalizations appeared more beneficial in cases of underserved training data, it is not clear if they were sufficiently statistically significant to draw firm conclusions.”
    - Replaced this sentence: “The AUC metrics shown are averages of three trials.”
    - With the following: “The AUC metrics shown are averages of repetition counts detailed in Appendix G.”

Best regards.

---

### Comment · ~Nicholas_Teague1 · 2021-01-14
**Final response**

Thank you reviewers. A few parting thoughts:

1. In addition to the other transformations surveyed, a key contribution here is the Automunge implementation of a push button solution to data augmentation by noise injection for tabular learning. This paper validated the numerical augmentations and found material benefit to model performance in cases of underserved training data, FYI subsequently we have performed additional experiments and found similar benefit of our approach to categoric augmentations.

2. It was disappointing that none of the reviewers understood the family tree primitives, which represent a fundamental reframing of command line specification of transformation sets as may include generations and branches of derivations. I believe this is a significant contribution and major differentiator for our library.

Best regards.

---

### Decision · Program_Chairs · 2021-01-07
**Final Decision**

**Decision:**

Reject

**Comment:**

While I'm sure there are many merits to the underlying work here, the consensus of the reviews is to recommend a rejection as an ICLR paper. That recommendation is based on issues with significance as well as on clarity issues, noted by reviewers even after the revisions.

One pattern I noticed was that it seemed unclear whether the paper was to be regarded primarily as a software paper or as a paper on preprocessing. Most initial reviews evaluated it primarily as a software paper, but some comments from the authors in the discussion period seemed to frame it instead as a paper about research on preprocessing (independent of software). See my other comment for more detail on this question.

Regardless of the intended framing, on significance as an ICLR submission, reviewers did not support its acceptance by either standard:
* R1 post-response: "Regarding how to frame the paper (either about feature pre-processing or the software library), my (favourable) interpretation is to frame it as about the software library. As a paper about feature pre-processing it would have even less merit."
* R3 post-response: "the work has more upside in the software contribution than the feature pre-processing research"
* R4 post-respones: "After reading the revised version and the author response, I am still not convinced that the paper makes a substantial contribution either on the fundamental research angle or the software library angle"

One specific issue was that the experiments did not adequately support the main claims:
* R2 post-response: "I feel that the experiments in Section 8 are still too limited to demonstrate the value of the techniques and software"
* R4 post-response: "While the main contributions of the paper are still a little ambiguous, the experiments don't seem to support the claims (ease of use of the library). The results seem to suggest that one of the contributions is the improved accuracy of results, but then the experimentation is far too limited to draw such a conclusion."

On clarity:
* R1 post-response: "The new section 5 is still very hard to understand and I still couldn't make sense of the family tree primitive mechanism"
* R3 post-response: "I agree with AnonReviewer1 about the difficulty of understanding the family tree primitives, both before and after the revision"
* R2 post-response: "I am still unclear on the family tree primitives [...] it's not clear to me what problem the family tree primitives solve"


The authors showed a lot of enthusiasm and good spirits in working to improve the submission. I hope the feedback provided here is useful.

However, based on the consensus of the reviews, I recommend rejection of this submission.